 eLIFE

# Functionally defined white matter of the macaque monkey brain reveals a dorso-ventral attention network

**Ilaria Sani[1]\*, Brent C McPherson[2], Heiko Stemmann[3], Franco Pestilli[2†], Winrich A Freiwald[1†]\***

[1]Laboratory of Neural Systems, The Rockefeller University, New York, United States; [2]Department of Psychological and Brain Sciences, Indiana University, Bloomington, United States; [3]Institute for Brain Research and Center for Advanced Imaging, University of Bremen, Bremen, Germany

**Abstract** Classical studies of attention have identified areas of parietal and frontal cortex as sources of attentional control. Recently, a ventral region in the macaque temporal cortex, the posterior infero-temporal dorsal area PITd, has been suggested as a third attentional control area. This raises the question of whether and how spatially distant areas coordinate a joint focus of attention. Here we tested the hypothesis that parieto-frontal attention areas and PITd are directly interconnected. By combining functional MRI with ex-vivo high-resolution diffusion MRI, we found that PITd and dorsal attention areas are all directly connected through three specific fascicles. These results ascribe a new function, the communication of attention signals, to two known fiber-bundles, highlight the importance of vertical interactions across the two visual streams, and imply that the control of endogenous attention, hitherto thought to reside in macaque dorsal cortical areas, is exerted by a dorso-ventral network.

DOI: https://doi.org/10.7554/eLife.40520.001

\*For correspondence:
isani01@rockefeller.edu (IS);
wfreiwald@rockefeller.edu (WAF)

†These authors contributed equally to this work

**Competing interests:** The authors declare that no competing interests exist.

## Introduction

Attention is the neuro-cognitive function that selects currently relevant pieces of information at the expense of irrelevant ones (*Carrasco, 2011*; *Chelazzi et al., 2011*). Attentional selection results from the interplay between exogenous factors, like intrinsic object saliency, and endogenous processes, like the relevance of an object for the task at hand. Endogenous attention is thought to be controlled by a fronto-parietal network (*Corbetta et al., 2008*; *Kastner and Ungerleider, 2000*). In the macaque monkey brain, the lateral intraparietal area (LIP; *Bisley and Goldberg, 2010*; *Gottlieb et al., 1998*) and the frontal eye field (FEF; *Thompson and Bichot, 2005*) exhibit key properties of attentional control and are thought to form the core of the endogenous attention network (*Stemmann and Freiwald, 2016*).

The fronto-parietal theory of attentional control is supported by recent whole-brain functional magnetic resonance imaging (fMRI) studies in the macaque monkey using a wide range of attention tasks (*Caspari et al., 2015*; *Patel et al., 2015*; *Stemmann and Freiwald, 2016*). One of these studies (*Stemmann and Freiwald, 2016*), found that an area in the lower bank of the Superior Temporal Sulcus (STS), the dorsal posterior infero-temporal area, PITd (*Felleman and Van Essen, 1991*; *Saleem and Logothetis, 2007*) was even more strongly attention-modulated than LIP, was engaged by multiple attention tasks, and was not modulated by the task-relevant feature dimension. Thus PITd was shown to exhibit key properties of an attentional control area (*Figure 1A–B*; *Stemmann and Freiwald, 2016*). These results raised the possibility that the network for

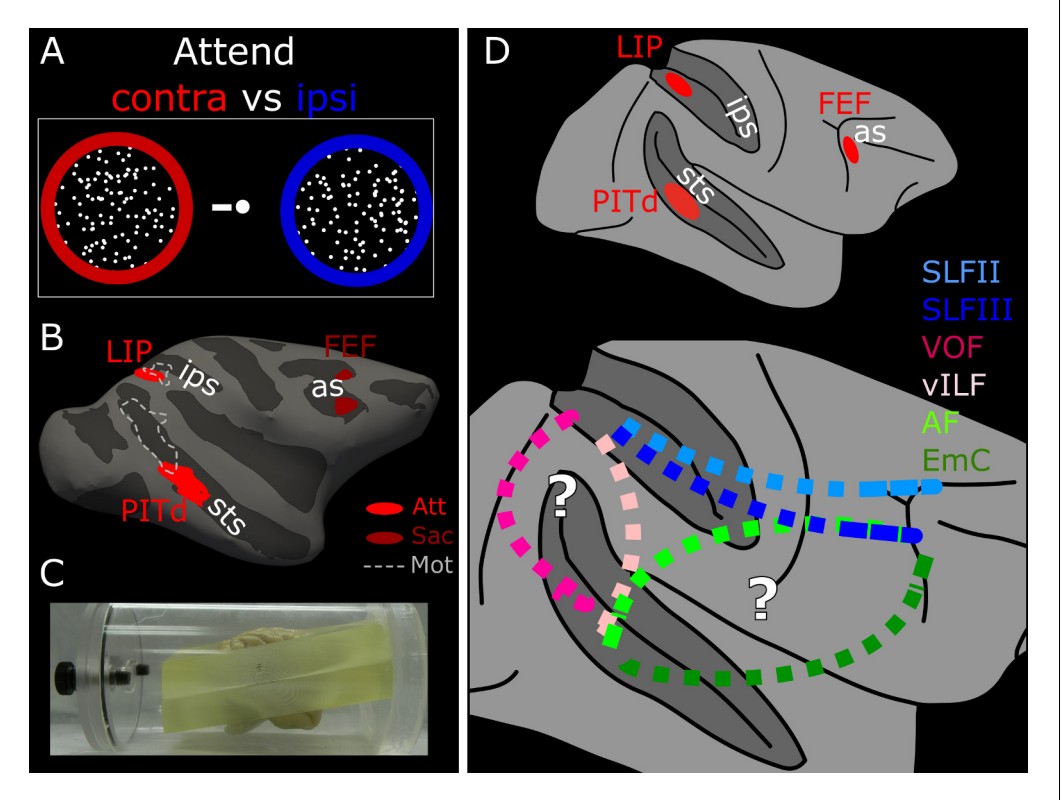

**Figure 1.** Functionally guided identification of the putative endogenous attention network. (A) Schematic of the two attentional conditions of the motion discrimination task performed during functional scanning. Monkeys alternated between paying attention to the right, passive fixation, and paying attention to the left. During the 'right' and 'left' blocks, subjects had to detect and discriminate a motion event at the cued location, while ignoring similar visual stimulation at the irrelevant location. (B) Schematic of the ROIs defined by attentional modulation, motion selectivity, and saccade modulation overlaid on the inflated right hemisphere of Monkey 1 (M1). PITd and LIP (light red) were significantly more activated by performance of the attentive motion-discrimination task on the right as compared to the left hemifield; FEF (dark red) was significantly more activate by performance of the saccade task as compared to fixation; dashed-line areas were significantly more activated during passive fixation of moving stimuli as compared to static stimuli. (C) Custom-made brain mold used to perform ex-vivo diffusion imaging. The device holds the brain with the anterior and posterior commissure aligned to the bore. (D) Whole brain model of the attention network of the Macaque as defined by functional activation in (*Stemmann and Freiwald, 2016*) (top). Schematic of putative connections and the hypothetical pathways between attention nodes PITd, LIP and FEF (bottom). *sts*, superior temporal sulcus; *ips*, intraparietal sulcus; *as*, arcuate sulcus; PITd, Posterior Infero-Temporal dorsal area; LIP, Lateral Intraparietal area; FEF, Frontal Eye Field; AF, Arcuate Fasciculus; EmC: Extreme Capsule; vILF: vertical branch of the Inferior Longitudinal Fasciculus; SLF: Superior Longitudinal Fasciculus; VOF, Vertical Occipital Fasciculus. See also *Figure 1—figure supplement 1*, and *Supplementary file 1*.

DOI: https://doi.org/10.7554/eLife.40520.002

The following figure supplement is available for figure 1:

**Figure supplement 1.** Functional cortical ROIs.

DOI: https://doi.org/10.7554/eLife.40520.003

endogenous control of attention may not be confined to fronto-parietal (dorsal) structures, but also includes a ventral region in the temporal lobe.

A ventral attention network has been previously proposed in humans (*Corbetta et al., 2008*; *Geng and Vossel, 2013*). However, its functional properties are profoundly different from those of macaque area PITd. First, human dorsal and ventral attention networks have a different functional profile, with the first primarily supporting endogenous attention, and the latter primarily supporting exogenous attention (*Corbetta et al., 2008*), along with context updating (*Geng and Vossel, 2013*) and social cognition (*Carter and Huettel, 2013*; *Mars et al., 2013*; *Schwiedrzik et al., 2015*). On the contrary ventral area PITd shared similar functional properties with dorsal areas LIP and FEF (*Stemmann and Freiwald, 2016*). Second, the most prominent part of the human ventral attention network, the temporo-parietal-junction (TPJ), is suppressed during endogenous attention, and

activated during target detection and shifts of attention, particularly when stimuli are salient or unexpected (*Corbetta et al., 2008*; *Corbetta and Shulman, 2002*), whereas PITd was strongly activated during prolonged sustained attention (*Stemmann and Freiwald, 2016*). Third, macaque PITd possesses a putative human homolog, phPITd, an area located far ventrally to TPJ (*Glasser et al., 2016*; *Kolster et al., 2010*). Thus the discovery of attention area PITd located ventrally to the ventral attention network raises important questions about attentional control mechanisms and their support circuitry.

In the brain imaging literature, the term network is often used loosely to describe a set of co-activated brain regions without explicit reference to their physical connections (*Petersen and Sporns, 2015*). The fact that multiple brain areas exhibit similar functional profiles across visual and cognitive conditions, like PITd, LIP and FEF did in (*Stemmann and Freiwald, 2016*), could be for a number of fundamentally different reasons. Co-activation might be initiated by a common driver outside the brain, like the visual stimulus, it could result from a common source inside the brain, with functional nodes only indirectly connected, or from connections that directly link the areas of interest. Only the characterization of the connectivity pattern between functionally identified areas establishes true network status. Understanding the structure of connectivity between putative attention control areas would thus be critical to establish whether they form an attention control network and help to answer the question of how they might coordinate a common focus of attention.

Long-range inter-areal connectivity is supported by the white matter. Uncovering the white matter tracts supporting endpoint-to-endpoint connections is important to know for at least two reasons. First, fascicles respond to behavior by adapting density, shape, and molecular composition in a manner that corresponds to individual abilities in health and disease (*Fields, 2008*; *Jbabdi et al., 2015*; *Pestilli, 2018*; *Rokem et al., 2017*; *Thomason and Thompson, 2011*; *Wandell, 2016*), and thus take an active role in shaping network function. To fully understand brain functionality, we thus need to combine knowledge about brain function and structure. Second, connectivity is crucial to elucidating how neurons and neural networks process information. The main organizing principle of the visual system is its separation into two major posterior-anterior information-processing streams: a dorsal path for spatial processing, and a ventral path for feature and object processing (*Kravitz et al., 2013*; *Milner and Goodale, 1995*). Each pathway may contain multiple, largely parallel routes (*Kravitz et al., 2013*), a conclusion that seems to be confirmed by the existence of dorsal and ventral attention areas. In this view, the study of connectivity between parietal and temporal attention nodes can serve as a model system for a general understanding of information flow between the dorsal and ventral visual pathways and between the parietal and temporal lobes at large.

Here we sought to, first, determine connections between attention areas PITd, LIP, and FEF, and, second, to delineate the long association fascicles of this postulated endogenous attention control network. Previous anatomical studies have demonstrated strong and direct connections between parietal area LIP and frontal area FEF (*Blatt et al., 1990*; *Lewis and Van Essen, 2000*; *Petrides and Pandya, 2006*) mainly via the Superior Longitudinal Fasciculus (SLF). However, function and anatomy of PITd have, so far, received little attention. More specifically, to date only a single classical anatomical study has targeted PITd (*Distler et al., 1993*), but due to the study's paucity of data and lack of functional characterization, the relationship to attention networks remains unclear. We thus asked whether PITd connects to LIP, whether it connects to the FEF, and which fiber pathways these cortical areas use to exchange attentional information (*Figure 1D*). The general anatomy of macaque white matter organization suggests different candidate pathways: PITd may connect to LIP through the Vertical Occipital Fasciculus (VOF), a fascicle that has been shown to have some ventral terminations in the temporal cortex (*Takemura et al., 2017*). Alternatively, PITd and LIP might connect through a more anterior fascicle, the vertical branch of the Inferior Longitudinal Fasciculus (vILF). Similarly, it would be possible for PITd and the FEF to connect via one of two major pathways: fibers might run via the Arcuate Fasciculus (AF), as areas in the caudal superior temporal regions do (*Petrides and Pandya, 2006*). Alternatively, fibers connecting PITd and FEF could course via the Extreme Capsule (EmC), as the middle superior temporal region and some anterior patches of the inferior temporal sulcus do (*Petrides and Pandya, 1988*; *Petrides and Pandya, 2006*).

To test these hypotheses, we developed a new approach that combines in-vivo fMRI with high-resolution ex-vivo diffusion imaging (dMRI) and ensemble probabilistic tractography. This experimental approach was specifically designed to facilitate the comparison of dMRI with classical tracer

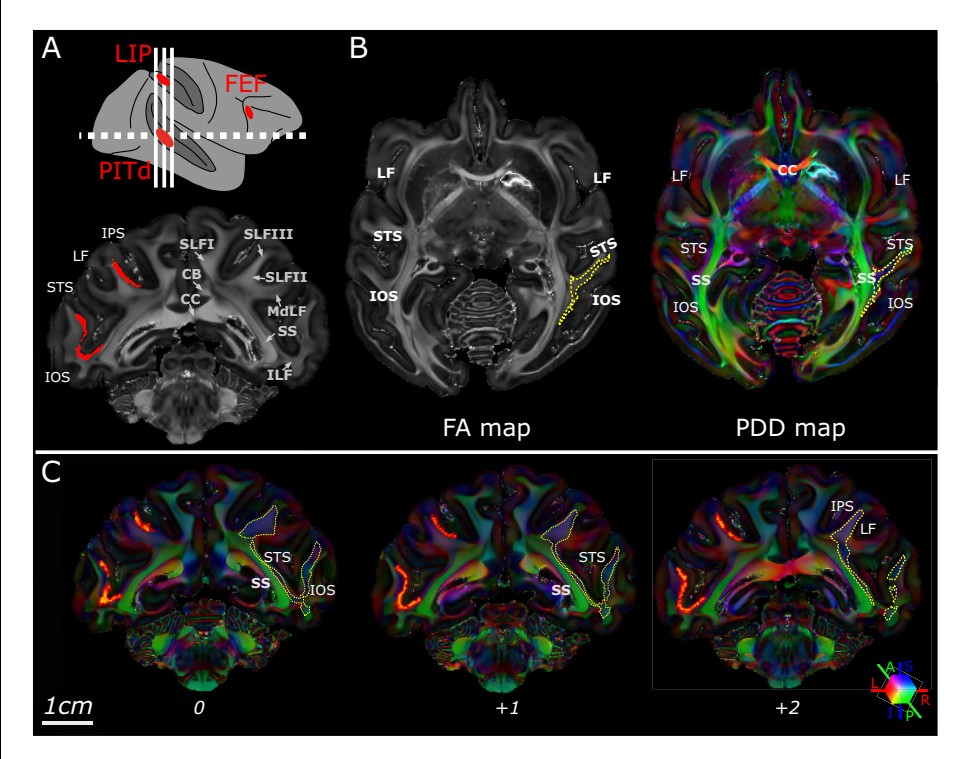

**Figure 2.** Macaque dorso-ventral attentional connections identified using Principal Diffusion Directions. Schematic of the macaque attention network identified with fMRI (top panel); the dashed white line indicates the section plane of axial images in B; white solid line indicates section planes of coronal images in A and C. Bottom panel shows a coronal-plane image of the Fractional Anisotropy (FA) map of subject M1; bright voxels in the FA map indicate anisotropic voxels. Note the clear separation between different association fascicles, labelled in the right hemisphere. CB: Cingulum Bundle CC: corpus callosum; MdLF: Middle Longitudinal Fasciculus; ILF: Inferior Longitudinal Fasciculus; SLF: Superior Longitudinal Fasciculus (branch I, II, III); SS: sagittal stratum. In the left hemisphere, original functional ROIs have been projected at the intersection between gray and white matter for PITd and LIP (red areas). ROIs have been then inflated, and used as seeds for tractography; (**B**) Axial-plane images of the FA map of subject M1 (left). Principal Diffusion Direction (PDD) map of subject M1 (right). Colors indicate the PDD in individual voxels; blue: superior-inferior, S-I; green: anterior-posterior, A-P; red: left-right, L-R; the position and properties of fascicle departing from PITd is visible in the white matter posterior to the STS (yellow dashed outlines were segmented automatically using the Matlab Image Processing Toolbox (RRID:SCR_001622), Color Thresholder (channel settings: R < 75; G < 75; B > 50). (**C**) Multiple coronal-plane images of the PDD map of subject M1. IOS: Inferior Occipital Sulcus; IPS: Inferior Parietal Sulcus; LF: Lateral Fissure; STS: Superior Temporal Sulcus. See also *Figure 2—figure supplement 1*.

DOI: https://doi.org/10.7554/eLife.40520.004

The following figure supplement is available for figure 2:

**Figure supplement 1.** Functionally guided inspection of white matter between PITd and LIP.

DOI: https://doi.org/10.7554/eLife.40520.005

studies, and overcome some of the limitations of dMRI and tractography. For example, this approach reduces artifacts and distortions, improves robustness to crossing fibers, and advances anatomical accuracy. This technique allows for the study of the connectivity of functionally specific brain areas, the exploration of fine-grain structural connectivity, and the description of the pathways connecting the functional areas. These three aspects are key to the understanding of brain functional networks, and the lack of a method combining them all has prevented, so far, the full characterization of the macaque attention network. High-resolution dMRI, combined with functionally-defined tractography, also facilitates the comparison between human and non-human cognitive networks, helps define which circuitry in the macaque attention system can serve as an accurate model for humans, and bridges the gap between traditional tracer studies and dMRI in humans.

## Results

### Ventral cortical area PITd directly connects to the fronto-parietal attention network

To unravel whether PITd communicates with attention areas LIP and FEF in parietal and frontal cortex, we combined fMRI localization of attention-modulated areas PITd, LIP, and FEF with ex-vivo dMRI (*Figure 1A–C*). We acquired high angular and spatial resolution dMRI, and preprocessed diffusion images (Material and methods). We first estimated the voxel-wise microstructure of the brain tissue, computed Fractional Anisotropy (FA), Principal Diffusion Direction (PDD) maps (Material and methods), and identified the position of putative connections (*Pajevic and Pierpaoli, 1999*; *Takemura et al., 2017*; *Thiebaut de Schotten et al., 2012*; *Yeatman et al., 2013*). Second, we performed ROI-to-ROI probabilistic tractography to create a macrostructural model of brain pathways, and, third, we compared tracking results with previous histological studies.

Attentional activation was defined through a demanding attention task requiring motion discrimination and spatial selection (*Figure 1A*; see also *Stemmann and Freiwald, 2016*). Monkeys were cued to pay attention to one of two random dots surfaces both to detect and discriminate the occurrence of a coherent motion event (see Material and methods). To isolate cortical areas modulated specifically by endogenous attention and define LIP and PITd, we contrasted blocks with two spatial attention conditions: attend contralateral and attend ipsilateral (*Figure 1B*, *Figure 1—figure supplement 1*, and *Supplementary file 1*). These spatial attention conditions were dissociated from saccade planning, and largely separated from motion selectivity (*Figure 1B*, dashed lines). FEF was defined by means of a guided saccade task where monkeys were required, in different blocks, to either maintain fixation on a central target, or make saccades (*Figure 1B*, see also *Stemmann and Freiwald, 2016*).

We first established the position of putative connections. In high-resolution FA and PDD maps, the core of several major tracts within the white matter was clearly identifiable as a bright area where fibers run in the same direction (*Figure 2A*). Crucially, we found a major superior-inferior diffusion direction (blue) in the lateral temporal white matter close to PITd (*Figure 2B*). This is compatible with a direct vertical connection between PITd and LIP. The tract was identifiable from PDD maps in multiple slices between the two ROIs (*Figure 2C*) and consistently in all subjects (*Figure 2—figure supplement 1*).

To confirm the existence of a direct connection between LIP and PITd, we performed probabilistic tractography and defined streamlines (i.e. putative fiber trajectories reconstructed by the tractography algorithm) between the two ROIs (see Material and methods). We found a white matter tract connecting the two areas (*Figure 3B*). We observed the core of the tract at a consistent position for all subjects and hemispheres (*Figure 3—figure supplement 1*, yellow streamlines), thus validating the hypothesis of a direct vertical connection (*Figure 1D*, dark and light pink dashed lines).

Using a similar approach, we asked whether PITd connects to prefrontal attention and oculomotor area FEF (*Schall and Hanes, 1993*; *Thompson and Bichot, 2005*).We performed probabilistic tractography between PITd and FEF and successfully identified a white matter tract directly connecting PITd and FEF (*Figure 3C*). We observed the core of the tract for all subjects and hemispheres (*Figure 3—figure supplement 1*; cyan streamlines). The findings support the hypothesis (*Figure 1D*, dark and light green dashed lines) that a direct connection between PITd and FEF exists.

To quantify the physical properties of the two white matter tracts, we calculated the number of streamlines, the length, the density (number of streamlines per $mm^3$), and the volume (number of voxels touched by the streamlines in $mm^3$), of the connections (see Material and methods). Across subjects, LIP-PITd connections comprised an estimated 777 ± 300 streamlines and PITd-FEF connections 188.5 ± 77.8 streamlines. The fronto-temporal tract was on average longer than the temporo-parietal connection (*Figure 3D*, left panel), it had a lower density and similar volume (*Figure 3D*, middle panels). Next, we used the Linear Fascicle Evaluation (LiFE, *Caiafa and Pestilli, 2017*; *Pestilli et al., 2014*) to assess the statistical evidence (strength of evidence, see Material and methods) supporting the existence of the fascicles. Briefly, we used a virtual lesion method (*Honey and Sporns, 2008*; *Pestilli et al., 2014*; *Takemura et al., 2016a*) to compare the prediction accuracy on the diffusion signals between two connectome models, that is two comprehensive maps of neural connections in the brain. The first connectome model contains all streamlines (unlesioned), and the second connectome model contains all streamlines except the tract of interest (lesioned; see also

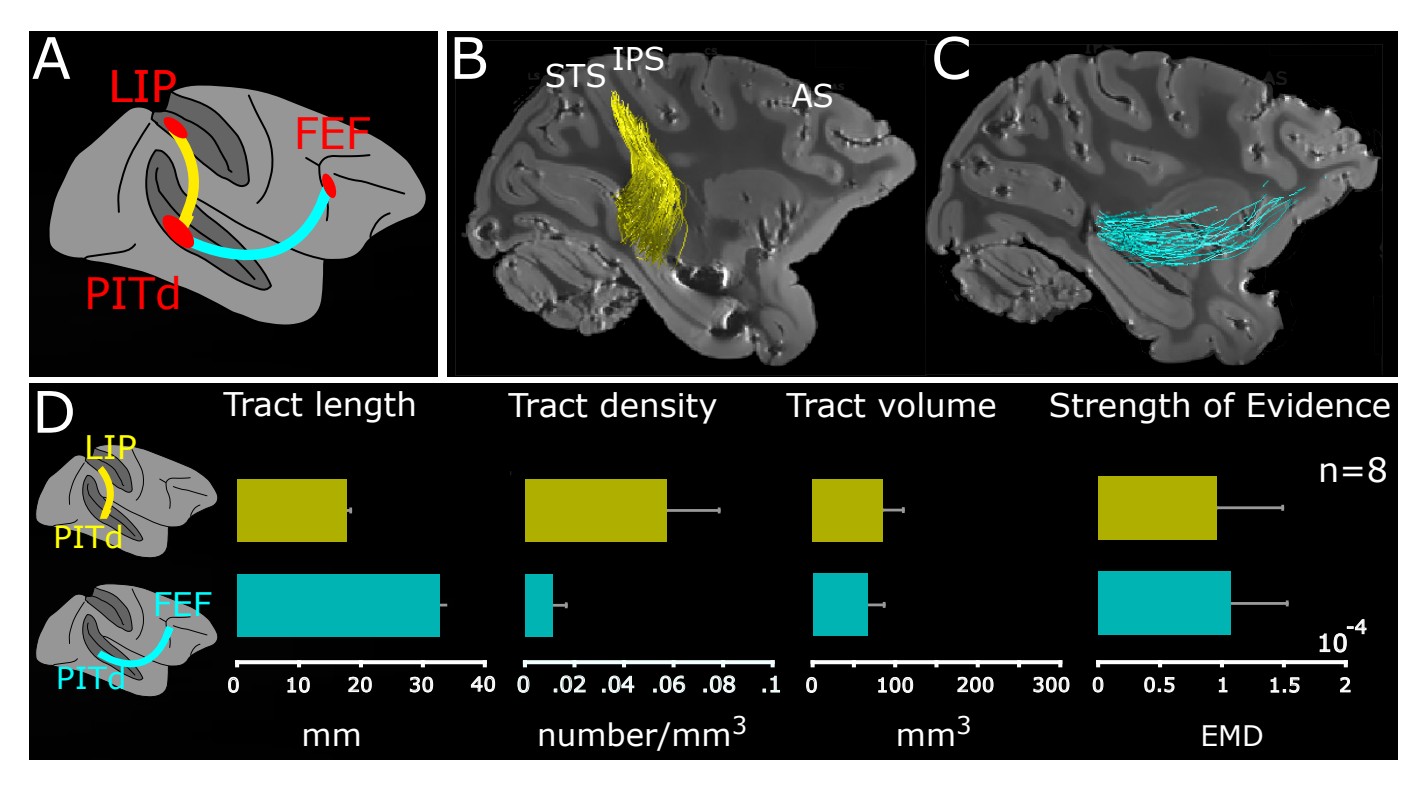

**Figure 3.** Macaque dorso-ventral attentional connection identified using tractography. (A) Schematic connections; PITd-LIP: yellow trace; PITd-FEF: cyan trace. (B-C) Sagittal-view of PITd-to-LIP and PITd-to-FEF connections overlaid on non-diffusion weighted (b = 0) image for M1. Conventions as in *Figure 1*. (D) Average tract length (mm); tract density (number of streamlines/mm³), tract volume (mm³) and strength of evidence are shown for functional tracts of all subjects. Error bars represent ±1 standard error of the mean across the eight hemispheres. See also *Figure 3—figure supplements 1*, *2* and *3*.

DOI: https://doi.org/10.7554/eLife.40520.006

The following figure supplements are available for figure 3:

**Figure supplement 1.** Tract properties are consistent across subjects and tracking methods.

DOI: https://doi.org/10.7554/eLife.40520.007

**Figure supplement 2.** quantitative anatomical properties.

DOI: https://doi.org/10.7554/eLife.40520.008

**Figure supplement 3.** Consistency across attentional ROI sets.

DOI: https://doi.org/10.7554/eLife.40520.009

Material and methods). The strength of evidence values demonstrate statistically that the measured data has signal consistent with the reported tracts (*Figure 3D*, right panel). In sum, the visible evidence on PDD maps (*Figure 2*), the quantitative descriptions of the streamlines (*Figure 3A–C*), the statistical evidence computed through LiFE (*Figure 3D*), all agree in supporting direct PITd-LIP and PITd-FEF wiring.

## vILF and EmC connect dorsal and ventral attention areas

Tractography is ideally suited to determine not only whether any two areas are connected to each other, but also which routes these connections take, a piece of knowledge typically not available in classical tracer studies (*Schmahmann and Pandya, 2006*). Thus next, we characterized the white matter tract anatomy of the macaque attention network. First, we tested whether the connections between functionally defined areas LIP and FEF conform to the previously established finding, that direct LIP-FEF connections run through a component of the Superior Longitudinal Fasciculus (SLF; *Figure 1D* dark and light blue dashed lines; *Blatt et al., 1990*; *Lewis and Van Essen, 2000*; *Petrides and Pandya, 2006*), a notion obtained without specifically targeting attentional areas.

Second, we tested the hypothesis that direct PITd-LIP connections are part of the vILF and distinct of the Vertical Occipital Fasciculus (VOF, *Figure 1D* dark and light pink lines), a fascicle that has been shown to have some ventral terminations in the temporal cortex and may therefore subserve PITd (*Takemura et al., 2017*). Third, we tested the hypothesis that the PITd-FEF pathway runs through the ILF and EmC, and is distinct from the Arcuate Fasciculus (AF; *Figure 1D* dark and light green lines; *Petrides and Pandya, 2006*). We tested these hypotheses in two ways. Firstly, we showed and quantified the spatial relationships between our functionally-defined bundles and major brain tracts. Using standard inclusion and exclusion anatomical planes (*Figure 4—figure supplement 1*; see also Material and methods) we defined the VOF, vILF, ILF, EmC, AF, SLFII and SLFIII, relying on anatomical information from previous tracer studies (for example, *Petrides and Pandya, 2006*; *Schmahmann and Pandya, 2006*; *Schmahmann et al., 2007*) as well as from dMRI studies (*Croxson et al., 2005*; *Eichert et al., 2018*; *Mars et al., 2016*; *Takemura et al., 2017*; *Thiebaut de Schotten et al., 2011*; *Thiebaut de Schotten et al., 2012*). We then established a direct comparison between our results and previous autoradiography studies (Material and methods), a histological approach able to show fiber path (*Schmahmann and Pandya, 2006*).

Probabilistic tractography between FEF and LIP demonstrated a direct pathway between these two dorsal attention areas (*Figure 4A* and *Figure 3—figure supplement 1*; orange). This fronto-parietal connection had an average length comparable to the PITd-FEF tract but showed higher density and volume (*Figure 3—figure supplement 2*). These results confirm numerous tracer studies reporting a strong connection between LIP and FEF. We next investigated whether the LIP-FEF connection may be part of SLF II or SLFIII, as defined in *Thiebaut de Schotten et al. (2011)* (see also *Figure 4—figure supplement 1A*). Consistent with previous autoradiography studies (*Schmahmann and Pandya, 2006*), SLFIII emerges from the inferior parietal lobule, courses horizontally through the white matter of the parietal and frontal operculum and reaches the region of the arcuate sulcus (*Figure 4A*, dark blue traces). Similarly, SLFII links the posterior parietal cortex to frontal lobe regions by running horizontally through the white matter dorsal to SLFIII (*Figure 4A*, light blue traces). The FEF-LIP tract is part of the most dorso-medial region of SLFIII and the most ventro-medial region of SLFII (*Figure 4B*, orange traces) thus lying at the intersection between SLFIII and SLFII and overlapping almost equally with both tracts (*Figure 4C*). These results complement previous studies by adding functional specificity and by localizing more precisely the anatomical position of FEF-LIP bundle.

To determine whether the PITd-LIP tract runs through the vILF or the VOF, we overlaid these two tracts and demonstrated their spatial separation (*Figure 4E*). LIP-PITd tract is located anterior to VOF, runs mainly vertically, but it is also elongated from posterior to anterior (*Figure 4F*, yellow streamlines) and it almost completely overlap with the vILF (*Figure 4G*). The VOF also runs vertically but lacks the anterior-posterior elongation and overall has more posterior endpoints which mostly serve occipital visual areas. We did not find any streamline running through the VOF (*Figure 4E*), and the only overlap between this tract and PITd-LIP connection occurs at the level of VOF termination in the temporal lobe (*Figure 4F–G*). Importantly, vILF doesn't seem to be easier to obtain when compared to VOF (*Figure 4H*). A control analyses, where both PITd-LIP tract and the VOF were functionally defined, confirmed the separation between the two bundles (*Figure 4—figure supplement 3*). Interestingly, when we functionally parceled areas nearby PITd to test for some degree of specificity in the way PITd connects to LIP, we found that PITd showed the highest degree of overlap with vILF (*Figure 4—figure supplement 4A–C*). We then directly compared this result with autoradiography studies and established that PITd-LIP connection is part of the vILF (*Figure 4F*, see also section 105 and 129 of *Schmahmann and Pandya, 2006*). Tracer studies have shown that fibers of the vILF terminated ventrally in areas V4t, MT, MST, and FST in the fundus of the STS and dorsally in the lower bank of the IPS (*Blatt et al., 1990*; *Schmahmann and Pandya, 2006*). Here we show that the vILF also contains a bundle connecting the two functionally defined attention areas PITd and LIP, and provide anatomical and functional evidence for the proposed distinction between VOF and vILF (*Takemura et al., 2017*).

To test whether PITd-FEF connection may be part of the horizontal limb of ILF and then enter the frontal lobe through the ventral component of the EmC, we defined the ILF as in *Takemura et al. (2017)* (see also *Figure 4—figure supplement 1C*), EmC as in *Mars et al. (2016)*, the AF as in *Eichert et al. (2018)*. ILF courses horizontally along the anterior-posterior axis of the temporal lobe within the white matter of the infero-temporal region and parahippocampal gyrus (*Figure 4I*, violet

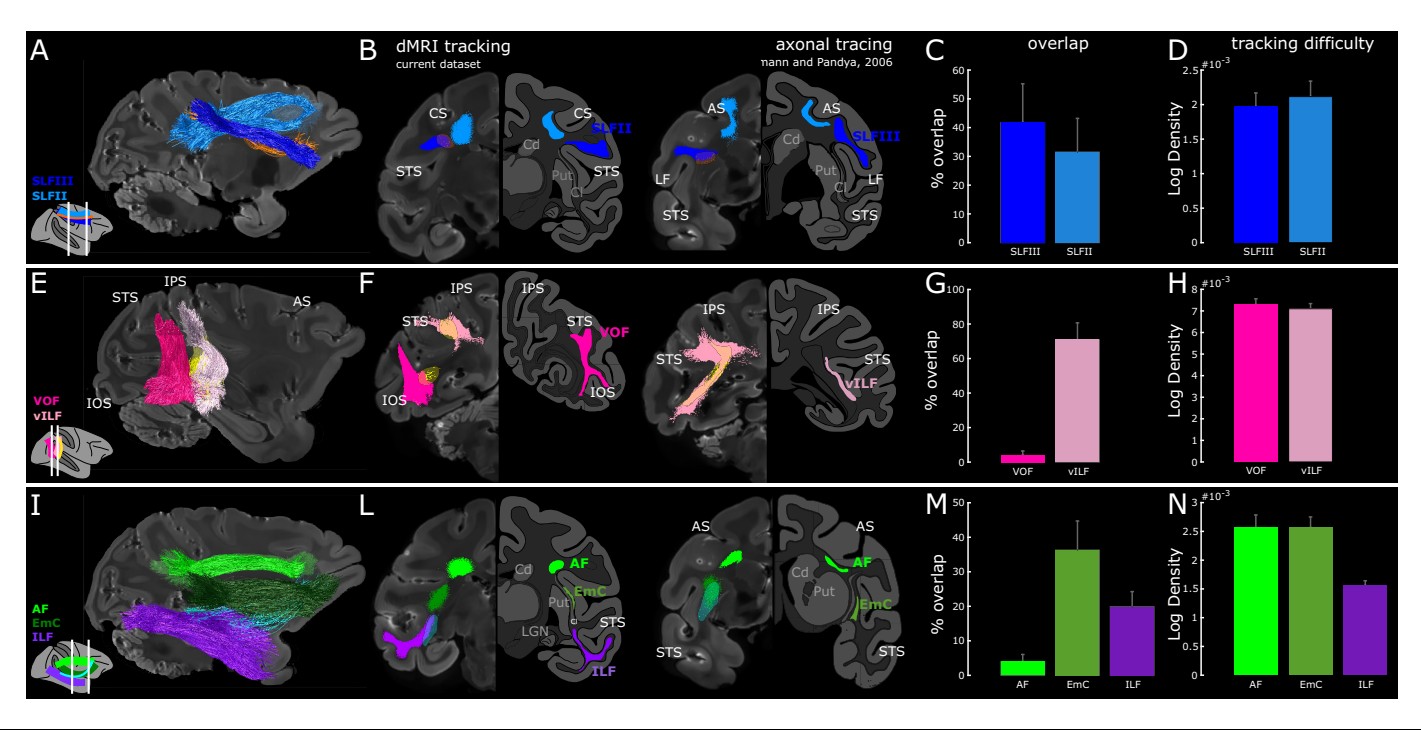

**Figure 4.** White matter within the vILF, EmC and SLF supports the endogenous attention network. (A) Sagittal view of SLF (dark and light blue) and LIP-FEF (orange) connectivity of subject M1. (B) Two coronal views of FEF-to-LIP connection and the SLF overlaid on the left hemispheres; the right hemispheres represent the schematic of the SLFII and the SLFIII (adapted from (*Schmahmann and Pandya, 2006*); sections 85,73, p. 542,540). (E) Sagittal view of the VOF, the vILF (dark and light pink), and LIP-PITd (yellow) connections of subject M1. (F) Two coronal views of PITd-to-LIP connection and the VOF overlaid on the left hemispheres; the right hemispheres represent the schematic of the vertical limb of the ILF (adapted from (*Schmahmann and Pandya, 2006*); section 129,105, p. 549,546). Note that VOF was recently described as a separate fasciculus in the monkey (*Takemura et al., 2017*) and previously called vILF in *Schmahmann and Pandya (2006)*; some inconsistency between tracer and diffusion literature, as well as between human and NHP literature still exists. While in the tracer literature the vertical connectivity in the posterior part of the brain is typically associated with vILF, in the diffusion imaging literature and especially in humans, several vertical tracts have been defined (see Discussion for more details). (I) Sagittal view of the ILF (violet), EmC (dark green), AF (light green), and PITd-FEF (cyan) connectivity of subject M1. (L) Two coronal-views of PITd-to-FEF connection and ILF; the right hemispheres represent the schematic of the horizontal limb of the ILF and of the Extreme Capsule (adapted from (*Schmahmann and Pandya, 2006*); section 85,57, p. 542,538). (C,G,M) Quantitative overlap between functionally defined attentional tracts and hypothesized anatomical pathways. (D,H,N) Tracking difficulty as measured through tract density (streamlines/mm³). Cd: Caudate nucleus; Cl: claustrum; LGN: Lateral Geniculate Nucleus; Put: Putamen. Other conventions as in *Figures 1–2*. See also *Figure 4—figure supplements 1*, *2*, *3* and *4*.

DOI: https://doi.org/10.7554/eLife.40520.010

The following figure supplements are available for figure 4:

**Figure supplement 1.** Anatomical waypoint ROIs.

DOI: https://doi.org/10.7554/eLife.40520.011

**Figure supplement 2.** Tract properties are consistent across subjects and tracking methods.

DOI: https://doi.org/10.7554/eLife.40520.012

**Figure supplement 3.** The VOF identified through ROI-to-ROI and waypoint tracking is separated from PITd-LIP tract.

DOI: https://doi.org/10.7554/eLife.40520.013

**Figure supplement 4.** Connectivity fingerprints of posterior STS areas.

DOI: https://doi.org/10.7554/eLife.40520.014

streamlines). ILF continues anteriorly within the temporal lobe up to the anterior end of the lateral geniculate nucleus, whereas FEF-PITd tract defined through functional ROI tracking departs from the ILF (*Figure 4I*, cyan streamlines) and joins the EmC (*Figure 4I*, dark green streamlines). A direct comparison of this results with autoradiography studies suggests FEF-PITd bundle leaves the seeded temporal ROI by travelling in the horizontal limb of ILF (*Figure 4L*, left panel) and then enters the frontal lobe through the ventral component of the EmC (*Figure 4L*, right panel). We also tested

whether the tract overlapped with AF (*Figure 4M*). We did not find any streamline running through the AF (*Figure 4I–L*, light green), and the only overlap between this tract and PITd-FEF connection occurs at the level of the termination in the frontal lobe. Importantly, EmC and ILF don't seem to be easier to obtain when compared to the alternative hypothesis AF (*Figure 4N*). Interestingly, we functionally parceled areas nearby PITd to test for some degree of specificity in the way PITd connect to FEF (*Figure 4—figure supplement 4D–F*). We found that PITd showed a similar degree of overlap with ILF/EmC as neighboring areas in the lower bank of the STS, and slightly higher overlap when compared to areas in the STS fundus, which are specialized for motion processing (see also Discussion). Thus, here we show that attention-modulated area PITd connects to the FEF through the ILF and the EmC, rather than through the AF. This wiring resembles that of middle superior temporal regions connecting to the prearcuate gyrus (*Petrides and Pandya, 1988*; *Petrides and Pandya, 2006*).

Collectively, these results thus establish that the vILF, but not the VOF, connects attention-modulated PITd with LIP, and that the EmC, but not the AF, connects PITd to FEF.

## Discussion

Endogenous attentional control in both humans and macaque monkeys is thought to be exerted by a fronto-parietal network (*Corbetta et al., 2008*; *Gottlieb et al., 1998*; *Kastner and Ungerleider, 2000*; *Thompson and Bichot, 2005*). Here we show that, in the macaque brain, classical attentional control areas LIP, in posterior parietal cortex, and FEF, in prefrontal cortex, are connected to PITd, an area in temporal cortex only recently implied in endogenous attention (*Stemmann and Freiwald, 2016*). We thus provide new evidence that PITd might be an endogenous attention control area, that attentional control may not be limited to dorsal fronto-parietal structures in the macaque brain, and that these areas form an attention network through direct connections (*Figure 5* and *Figure 5—video 1*). The fine anatomical characterization of the bundles connecting the three areas constitutes the first connectivity model of the macaque endogenous attention and should inform future theories of attentional processing.

PITd is a visual area located in the lower bank of the STS, it contains neurons that are shape- and color-selective (*Conway and Tsao, 2009*; *Hikosaka, 1998*) and have circumscribed receptive fields slightly larger than those of V4 (*Hikosaka, 1998*). Its anatomical connectivity and its role in cognition have been little explored. While LIP-FEF connectivity has been established through tracer studies

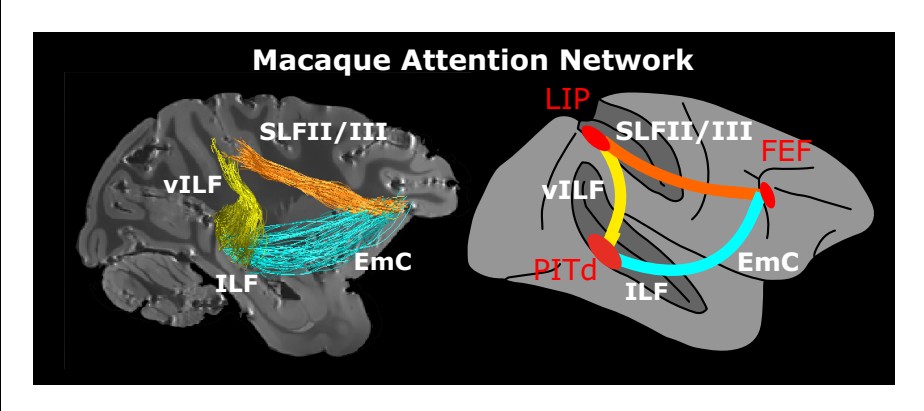

**Figure 5.** Macaque endogenous attention network. Whole brain model and structural connectivity of the attention network of the macaque as defined by functional activation (red areas, *Stemmann and Freiwald, 2016*) and structural connectivity. Colored solid lines represent connections and pathways between attention nodes PITd, LIP and FEF. Conventions as in *Figure 1*. See also *Figure 5—video 1*.

DOI: https://doi.org/10.7554/eLife.40520.015

The following video is available for figure 5:

**Figure 5—video 1.** Macaque dorso-ventral attention network.

DOI: https://doi.org/10.7554/eLife.40520.016

(*Blatt et al., 1990*; *Lewis and Van Essen, 2000*; *Petrides and Pandya, 2006*), connectivity of either area with the attention-modulated part of PITd has hitherto remained unknown. Even general connectivity with PITd, anatomically defined, was not fully established. One previous anatomical study investigated cortical connections of inferior temporal areas (*Distler et al., 1993*) and, through a single tracer injection in PITd, provided some evidence for connectivity between PITd and LIP. Due to the study's paucity of data and lack of relating connectivity to functional characterization, the relationship to attentional control networks remained unclear. Similarly, tracer studies targeting LIP and FEF showed variable results in terms of existence and strength of connections with PITd (*Barbas and Mesulam, 1981*; *Barbas, 1988*; *Blatt et al., 1990*; *Bullier et al., 1996*; *Distler et al., 1993*; *Schall et al., 1995*). These inconsistencies are likely the result of differences in tracing methods, injection sites, anatomical definitions, and possibly even nomenclature (*Felleman and Van Essen, 1991*; *Kötter, 2004*; *Stephan et al., 2001*). We resolved these difficulties by combining functional mapping with dMRI. The part of PITd that is strongly modulated by attention in ways similar to those of LIP and FEF, is directly connected to these two classical attentional control regions. The comparative tractography we conducted between PITd and nearby functional areas (*Figure 4—figure supplement 4*) confirms the existence of tract-specific connections from PITd and demonstrates that, structurally, this connectivity is part of a larger-scale pattern of connectivity that PITd shares with its immediate neighborhood. Yet, functionally, PITd differs fundamentally from nearby STS areas in not being motion selective, while being very strongly attention modulated during attentive motion discrimination. Thus, functional specificity of PITd does not arise as a simple consequence of its connectivity with LIP and FEF. It is noteworthy that this connectivity scheme is different from the nearby face-processing network (*Grimaldi et al., 2016*; *Moeller et al., 2008*), which encompasses two face areas right next to attention-modulated area PITd (*Stemmann and Freiwald, 2016*). In the face-processing network, tight connectivity between face areas helps to ensure domain specificity (for faces) (*Grimaldi et al., 2016*; *Moeller et al., 2008*). Yet, for attention control networks shared connectivity fulfills the need to exert control over multiple areas and gather visual information from across multiple visual domains (*Fecteau and Munoz, 2006*). In particular, PITd's position within the ventral visual pathway is likely crucial to collect attention-relevant information about properties of objects (*Conway and Tsao, 2009*; *Hikosaka, 1998*), including faces (*Tsao et al., 2006*) and gaze (*Marciniak et al., 2014*), possibly providing information on object value (*Ghazizadeh et al., 2018*). Through the described connections, PITd might thus make important contributions to core functionality of endogenous attention and to a wide range of attention-related functions.

In humans, attention has been proposed to be controlled by two networks with different functions, located in the dorsal and ventral part of the brain (*Corbetta et al., 2008*). The most prominent part of the ventral attention network is the temporo-parietal-junction (TPJ). TPJ is suppressed during endogenous attention, and activated during target detection and shifts of attention, particularly when stimuli are salient or unexpected (*Corbetta et al., 2008*; *Corbetta and Shulman, 2002*). It is thus thought to be functionally, but also anatomically, distinct from the dorsal fronto-parietal network (*Corbetta et al., 2008*). An area homologous to TPJ in the macaque has been hypothesized, but not yet found (*Patel et al., 2015*). Macaque PITd is *activated* during sustained attention (*Stemmann and Freiwald, 2016*), does not activate during attentional shift (*Caspari et al., 2015*), and, as we show here, is structurally connected to the dorsal fronto-parietal network. PITd thus bears neither functional nor structural similarity to human TPJ. One of the macaque candidate regions previously suggested for the putative correspondence to the human TPJ is the dorsal bank of STS and the adjacent superior temporal gyrus. Lesions of this region cause neglect-like (or spatial extinction-like) syndromes (*Luh et al., 1986*; *Scalaidhe et al., 1995*; *Watson et al., 1994*), which led to a suggestion of a possible homology to the (right hemisphere) superior temporal gyrus (STG) in humans, one of the major lesion foci in neglect patients (*Karnath, 2001*). Alternatively, macaque TPO/STP and neighboring parietal area seven were hypothesized to correspond to human TPJ. These dorsal STS areas show, similarly to areas LIP and FEF, memory delay activity during saccade preparation (*Kagan et al., 2010*), and exhibit task-specific changes after reversible LIP inactivation (*Wilke et al., 2012*). Furthermore, recent work on the social brain also suggested that a potential precursor to human TPJ may reside in macaque posterior dorsal STS (*Schwiedrzik et al., 2015*) or in area 7a (*Sliwa and Freiwald, 2017*). Furthermore, macaque PITd possesses a putative human homolog, phPITd, an area located far ventrally to TPJ (*Glasser et al., 2016*; *Kolster et al., 2010*). Human phPITd occupies a similar positions in retinotopic cortex as macaque PITd, and, like PITd, is sensitive

to two-dimensional shapes and is insensitive to motion (*Kolster et al., 2010*). Our results predict, under a homology argument of attention network organization, this area as the ventral node of the human endogenous attention network.

PITd connects to LIP and FEF through specific white matter tracts. Knowing these tracts is important for a full characterization of endogenous attentional control circuits, their vulnerabilities, and developmental potential. For example, the properties of specific fiber-bundles have been found to correlate with attentive behaviors (*Chechlacz et al., 2015*; *Chica et al., 2018*; *Parlatini et al., 2017*), and their disconnection results in attentional deficits like spatial neglect (*Carter et al., 2017*; *Lunven and Bartolomeo, 2017*), it is therefore of great importance to characterize their structure and path. Traditional tracer studies typically focus on the analysis of the origins and terminations of white matter fibers, without characterizing the white matter pathways between the endpoints. Our experimental approach, by capitalizing on the macroscopic bird-eye view of dMRI (*Jbabdi et al., 2015*; *Rokem et al., 2017*) combined with behaviorally guided fMRI (*Stemmann and Freiwald, 2016*), allowed us also to describe the cortico-cortical *pathways* of the endogenous control attention. We established that the vertical branch of the ILF, but not the VOF, connects the two posterior attention areas PITd with LIP, and that the ILF and the EmC, but not the AF, connects PITd to the dorsal prefrontal area FEF. Our results thus imply a role in attentional control for the vILF and the ILF/EmC. (*Figure 5* and *Figure 5—video 1*).

We have used quality data, combined multimodal data, and used advanced quantification and statistical evaluation methods (see Material and methods), yet we fully acknowledge that tractography has limitations and is challenged by the trade-off between specificity and sensitivity (*Thomas et al., 2014*). Recent studies have demonstrated limitations of dMRI-based tractography in three important way: (1) in the ability to solve crossing fibers (*Maier-Hein et al., 2017*; *Roebroeck et al., 2008*), (2) in the dependency on tractography algorithms or parameter settings (*Bastiani et al., 2012*; *Chamberland et al., 2014*; *Domin et al., 2014*; *Kunimatsu et al., 2004*; *Parizel et al., 2007*; *Takemura et al., 2016a*; *Taoka et al., 2009*; *Thomas et al., 2014*), and (3) in the accuracy for estimating fiber projections into cortical gray matter (*Reveley et al., 2015*). Our results set the foundation for future investigations using other modalities to infer the connectivity of attentional control circuitry in macaque monkeys and other species, like human subjects.

In humans, two distinct vertical fiber bundles that connect the dorsal and ventral visual streams have been described, the VOF and the posterior Arcuate Fasciculus (pAF). The VOF supports a variety of perceptual functions, like skilled reading and spatial orienting (*Takemura et al., 2016b*; *Yeatman et al., 2014*; *Yeatman et al., 2013*), the pAF, located anterior to the VOF (*Catani and Thiebaut de Schotten, 2008*; *Thiebaut de Schotten et al., 2014*; *Weiner et al., 2017*), is thought to support language (*Catani and Thiebaut de Schotten, 2008*; *Thiebaut de Schotten et al., 2014*) and action-to-object coordination (*Budisavljevic et al., 2018*). The existence of two separated vertical tracts in the posterior part of the human brain resembles the connectivity profile of the monkey described here and suggest a possible resolution of current cross-species inconsistencies in the literature. Because of its anatomical location, its cortical endpoints, and its separation from the VOF, human pAF may be a candidate tract for the exchange of attentional information between dorsal and ventral attention areas in the posterior part of the human brain, thus being functionally and anatomically similar to the vILF. The fronto-temporal connection between macaque PITd and FEF via the EmC we found in this study, may also have a human homolog. Human EmC is important for language (*Catani and Bambini, 2014*; *Friederici and Gierhan, 2013*) and may also carry attentional information (*Umarova et al., 2010*). Umarova and colleagues found that human EmC connects ventral frontal cortex and insula to the TPJ, namely the core nodes of the human exogenous attention network (*Corbetta et al., 2008*; *Corbetta and Shulman, 2002*). A direct connection via EmC between phPITd and FEF would suggest a division of function between phPITd and TPJ, but similar frontal lobe connectivity. Alternatively, the functional difference between PITd and TPJ may be paralleled by a difference in structural connectivity: the AF may be a good candidate to exchange endogenous attention information between human frontal and temporal lobes. Future work examining both structure and function in the human brain should be able clarify the structural similarity between macaque and human attention networks and thus reveal putative constraints by white-matter fascicles on the evolution of functional networks. Determining the mesoscopic correspondence between structure and function across species is a fundamental challenge for neuroscience (*Zhang et al., 2013*), a challenge that cannot be met by pure functional and pure anatomical

measurements alone (*Pollen and Hofmann, 2008*). The first fine-scale functional characterization of the white matter tracts supporting endogenous attention described here sets up a new framework to understand the evolution of function in white matter bundles, and to determine the relationship between brain structure and function during evolution.

The main organizing principle of the visual system is its separation into two major posterior-anterior information-processing streams: a dorsal path for spatial processing, and a ventral path for feature and object processing (*Kravitz et al., 2013*; *Milner and Goodale, 1995*). Separation of information into different streams, however, appears to be at odds with the requirements of visual attention which generally needs to integrate spatial and featural information (*Baylis et al., 2001*; *Milner, 2017*; *Treisman and Gelade, 1980*). One possible solution to this binding problem, is the development of shape-selectivity in the dorsal stream (*Theys et al., 2015*). Alternatively, vertical connections between the dorsal and ventral attention areas as described here, provide a structural substrate for attentional integration that capitalizes on the full functional repertoire of both streams. This vertical connectivity may thus be a general theme, a second organizing principle of the visual system in humans (*Kay and Yeatman, 2017*; *Pestilli et al., 2014*) and macaques (*Nelissen et al., 2011*), extending from attention to other functions requiring vertical integration like language (*Cloutman, 2013*) or skilled grasping (*van Polanen and Davare, 2015*).

Taken together, our work advances the understanding of the architecture of attentional control by suggesting a three-node cortical network, with nodes in three different cortical lobes as the structural substrate of this cognitive function. The description of direct structural connections between these nodes we provide here, now allows investigations of the 'effective connectivity' between attention areas and the causal impact of these interactions. The findings reported here have important consequences for the understanding of the mechanisms and functional capabilities of endogenous attention, for cognitive theories of endogenous attention (*Itti et al., 2005*), for the evolution of attention systems, and for the functional organization of the primate brain.

## Materials and methods

### Animal model and sample preparation

All animal procedures met the National Institutes of Health Guide for Care and Use of Laboratory Animals, and were approved by the local Institutional Animal Care and Use Committees of the Rockefeller University (protocol number 15849 hr). The brain of four adult male Rhesus macaque monkeys (*Macaca mulatta*, M1-4) (~10–15 y old,~9 kg) were used in this study. Animals were euthanized first by sedation using ketamine and dexdomitor, followed by deep anesthetization using pentobarbital and phenytoin, and then were perfused transcardially with 4% (wt/vol) paraformaldehyde in phosphate buffer. The brains were extracted and immersed for 24 hr in the same fixation solution. Gadolinium- diethylenetriamine penta-acetic acid (Gd-DTPA) (Magnevist, Berlex Laboratories) was added at 0.1% vol to the fixation solution. After a 24 hr fixation, the brain was soaked for 5–9 weeks at 4°C in phosphate-buffered Saline (PBS) mixed with 0.1% Gd-DTPA to reduce the longitudinal relaxation time (T1) (*D'Arceuil et al., 2007*; *Wedeen et al., 2008*). The brains were transferred from the Gd-DTPA– doped PBS into a sealable container built with a plexiglass tube (60 mm i.d.) for scanning. The container was filled with liquid Fomblin (Solvay Solexis) to match closely the susceptibility of the brain tissue. A custom-made brain mold held the brain securely in space, with the anterior and posterior commissure aligned to the bore (*Figure 1C*).

### Diffusion MRI data acquisition

A 7-Tesla Biospec 70/30 USR Avance III MRI scanner (Bruker BioSpin) equipped with 20 i.d. gradients with a strength of 450 mT/m and a rise time of 100 µs was used for imaging. Acquisition parameters were set as in *Thomas et al. (2014)*. In brief, diffusion data were acquired with a 3D spin-echo diffusion-weighted EPI sequence. b-value was set at 4,025 s/mm$^2$ (*Thomas et al., 2014*), which has been demonstrated to be sufficient to model multiple fiber populations in ex vivo specimens (*Dyrby et al., 2007*). Two diffusion-weighting gradient tables of 60 directions each was used, and ten additional image volumes were collected with b = 0 s/mm$^2$. Echo time was reduced to TE = 36.5 ms by acquiring the EPI train in 16 segments, and the repeat time was TR = 500 ms. The 3D image matrix size was 320 × 320 × 256, resulting in a near-isotropic spatial resolution of 250 × 250 × 254

μm. The entire DWI acquisition took ~74 hr. We also acquired two phase maps and two sets of reversed gradient EPI images with b = 0 s/mm2, one each before and one after the main DTI scan. Reversed gradient were used for distortion correction.

## Preprocessing, distortion correction, and Tissue Segmentation

The diffusion weighted images were corrected for motion, pulsation and eddy-current artifacts before image processing using TORTOISE (*Pierpaoli et al., 2010*), despite the amount of misregistration in the data was negligible on visual inspection. More specifically, we performed a two-step correction: DIFF_PREP - software by using a structural image as target, performed image resampling, motion, eddy current distortion, and EPI distortion correction, while DR-BUDDI- software by using pairs of diffusion data sets acquired with opposite phase encoding (blip-up blip-down acquisitions), performed additional EPI distortion correction. Images were additionally denoised by means of non-local mean algorithm (sigma = 0; dipy toolbox, https://github.com/nipy/dipy). Before applying statistical evaluation, voxels with extremely low signal were removed.

## Functional and anatomical MRI: Tissue segmentation

Attentional and visual regions of interests (ROIs) were defined functionally and anatomically. Two of the monkeys (M1 and M2) previously underwent whole brain functional imaging while performing a demanding attentional task. Experimental procedures are described in details elsewhere (*Stemmann and Freiwald, 2016*). In brief, the animals were cued to attend either the left or the right hemifield. They had to detect and discriminate the direction of motion of random dot stimuli at the cued location while ignoring a similar stimulus displayed in the opposite hemifield by making a saccade towards one of eight possible targets. Attentional ROIs were defined by contrasting blocks where the monkeys attended the ipsilateral hemifield versus blocks where the monkeys attended the contralateral hemifield (*Figure 1A–B*). The two attentional conditions were dissociated from saccade planning, since saccades to any of the eight different targets were generated equally frequently in the two blocks. Additional ROIs were defined by five localizer experiments, including retinotopic, motion and saccade mapping (*Supplementary file 1*). Functional ROIs maps used for tracking were created by projecting the volumetric functional ROIs at the intersection between gray and white matter on a T1 image (*Figure 1—figure supplement 1* and see also below).

To compare functional tracts with standard anatomical fascicles we defined inclusion and exclusion waypoints ROIs (*Figure 4—figure supplement 1*). The waypoint ROIs were defined in locations that isolate the central portion of the tract where the fibers are coherently bundled together and before they begin diverging towards cortex. Specifically, we defined the second and third branch of Superior Longitudinal Fascicle (SLFII and SLFIII) as in *Thiebaut de Schotten et al. (2011)*, the Inferior Longitudinal Fasciculus (ILF), the Extreme Capsule (EmC) as in *Mars et al. (2016)*, the Arcuate Fasciculus (AF) as in *Eichert et al. (2018)*, the vertical branch of the ILF (vILF) as in *Schmahmann and Pandya (2006)* and, the Ventral Occipital Fasciculus (VOF) as in *Takemura et al., 2017*. As an additional control, the VOF was defined also through ROI-to-ROI tracking. A previous study showed that most of the endpoints of the VOF were in V4d and V4v (*Takemura et al., 2017*). We functionally defined V4 through retinotopic mapping (*Supplementary file 1*; see also *Stemmann and Freiwald, 2016*), and used V4d and V4v as seed regions (*Figure 1—figure supplement 1*; *Figure 4—figure supplement 3*).

To compare the connectivity profile of PITd with that of nearby areas, we functionally and anatomically parceled areas nearby PITd and analyzed the connectivity of those regions with LIP and FEF. More specifically, we defined MST, MT, and FST by means of a motion localizer, V4t by means of attentional modulation, PL by using a face localizer, and TEa anatomically (*Figure 4—figure supplement 4A,D*; see also *Stemmann and Freiwald, 2016*). Then, we tracked the connections between these areas and functionally defined target attention areas LIP and FEF (see below).

The white and gray matter border was defined using a B0 image. An initial segmentation was performed using an automated procedure in MRtrix (erode/threshold, [*Tournier et al., 2012*] and refined manually [*Yushkevich et al., 2006*]; http://www.itksnap.org/pmwiki/pmwiki.php).

Functional ROIs maps used for tracking were created by projecting the volumetric functional ROIs at the intersection between gray and white matter on a T1 image (*Figure 1—figure supplement 1*). ROIs and the T1 were then aligned to the high-resolution B0 diffusion image (*Figure 2A*) using the

Advanced Normalization Toolbox (Ants, *Avants et al., 2011*).ROI projections were then smoothed and inflated to find their intersection with white matter tracts (FSL, *Jenkinson et al., 2012*). This method has been previously used in human (*Pestilli et al., 2014*; *Takemura et al., 2016b*; *Yeatman et al., 2014*). Attentional and visual ROIs were functionally mapped in two animals (M1 and M2) and directly used for tracking. Since M1 and M2 underwent both fMRI and electrophysiology investigations, M3 and M4 diffusion imaging data were acquired to control for the possibility that electrophysiological recordings near the functional ROIs of the two mapped brains may have introduced discontinuities in the white matter. For M3 and M4, attentional and visual ROIs from monkeys M1 and M2 were warped to the ex-vivo brains ROIs and the B0 of M1 and M2 were aligned to the high-resolution B0 diffusion image of M3 and M4 using the Advanced Normalization Toolbox (Ants). Both sets of ROIs (from M1 and M2) were tested and gave consistent results (*Figure 3—figure supplement 3*). Results mostly report data from M1 ROIs.

## Diffusion MRI data analyses

Our experimental approach was specifically designed to facilitate the comparison of dMRI with classical tracer studies and overcome some of the limitations of dMRI and tractography. We acquired data ex-vivo to eliminate experimental confounds and artifacts such as motion, noise, cardiac pulsation, and EPI distortion. We acquired high angular and spatial resolution dMRI, which –along with our relatively small ROIs - reduced the inherent coarseness of dMRI tractography (*Thomas et al., 2014*). We used a fine grained white matter mask to reduce the potential of tractography to introduce fiber continuity where there is none (*Basser and Pajevic, 2000*). We used probabilistic CSD tractography (MRtrix 0.2 - RRID:SCR_006971 (*Tournier et al., 2012*), see below) to improve the robustness of the tractography results, especially to crossing fibers (*Tournier et al., 2012*). We used ensemble tractography, that is a combination of candidate streamlines obtained by systematically varying tracking parameters (*Takemura et al., 2016a*; see also below) to be able to describe tracts that differ in length and curvature, to improve anatomical accuracy, and to reduce variations in the extent and strength of derived pathways, shown to be dependent upon parameters of the tractography algorithm (*Jones, 2010*; *Thomas et al., 2014*). We tracked each bundle multiple times to have an estimate of variability of the probabilistic procedure (*Figure 3—figure supplement 2A*). As a final step, we used a model-based approach to the selection of fibers which removes most false positive tracts (*Pestilli et al., 2014*). In brief, we estimated how much each fascicle in the candidate connectome contributes towards predicting the diffusion signal and used a virtual lesion method (*Honey and Sporns, 2008*; *Pestilli et al., 2014*) to characterize the strength of evidence supporting the fascicles of interest (Earth Mover's Distance, EMD [*Pestilli et al., 2014*; *Rubner et al., 2000*]; see also below).

## Tractography

Tracking of potential streamlines was performed using MRtrix 0.2 (*Tournier et al., 2012*). The white matter volume was used as seed region for connectome generation. The white matter volume and two ROIs of interests were used as seed region for single tracts of interest. We used constrained spherical deconvolution (CSD; *Tournier et al., 2007*) and probabilistic ensemble tractography (*Takemura et al., 2016a*) to reduce the possibility that tractography estimates would miss a real fascicle. Specifically, we used four curvature thresholds (minimum radius of curvature 1, 2, 3, 4) and four values of maximum number of harmonics (Lmax = 2, 4, 6, 8). We set other parameters as: step size 0.2 mm; maximum length 200 mm; minimum length 5 mm. For each tract and connectome, we generated 200,000 streamlines. We then merged the tracts and the connectomes.

We used two ROI-based methods to track: ROI-to-ROI and waypoints tracking. For each tract, the set of potential streamlines was refined by removing outliers on the basis of length and distance from the core portion of the tract (*Yeatman et al., 2013*). For each tract, the set of potential streamlines was refined by removing outliers. Specifically, for tracts defined with cortical ROIs we removed streamlines with length $\geq 4$ SD longer than the mean streamline length in the tract, and position $\geq 4$ SD away from the mean position of the tract. For tracts defined with waypoint ROIs we removed streamlines with length $\geq 3$ SD longer than the mean streamline length in the tract, and position $\geq 3.5$ SD away from the mean position of the tract (*Yeatman et al., 2012*).

To evaluate the statistical evidence supporting the existence of the fascicle, we used a Linear Fascicle Evaluation method (LiFE, *Pestilli et al., 2014*). We estimated how much each fascicle in the candidate connectome contributes towards predicting the diffusion signal and assigned a weight to each streamline. We then eliminated fascicles with zero weight to create an optimized connectome and optimized tracts. We used a virtual lesion method (*Honey and Sporns, 2008*; *Pestilli et al., 2014*) to characterize the strength of evidence supporting the fascicles of interest. We generated 'lesioned' connectome model by excluding the fascicle from the optimized connectome. We compared the prediction accuracy of lesioned model with that of optimized connectome ('unlesioned' model). We used the Earth Mover's Distance (EMD; *Pestilli et al., 2014*; *Rubner et al., 2000*) to estimate the strength of evidence.

To get a quantitative description of the tracts of interest we measured the number of streamlines, average tract length, tract volume (number of voxels touched by a streamline * voxel volume) and tract density (2 * number of streamlines / (number of voxels in both ROIs * voxel volume)). To measure how difficult the tracking process was for the different anatomical tracts being compared, we calculated tract density (*Figure 4D,H,N*), a measure that takes into account the number of streamlines obtained and corrects it for the ROI size. To quantify the similarity between attentional tracts and anatomical pathways, and therefore test our alternative hypotheses, we calculated the percentage of tract overlap as overlapping volume in the two tracts (unique streamline coordinates) divided by the total volume of the attention tract. To analyze the connectivity fingerprint of PITd and nearby functionally defined areas, we calculated the percentage of overlap of each single tract with traditional anatomical pathways (*Figure 4—figure supplement 4*, panels B,C,E,F). Tracts were overlaid on a non-diffusion weighted (b = 0) image and directly compared with matching anatomical slices obtained through autoradiography studies (*Schmahmann and Pandya, 2006*).

## Acknowledgements

We thank Eric Aronowitz and Henning U Voss for setting up diffusion MRI sequences; Margaret Fabiszak for help with sample preparation; James M Petrillo for his help in designing and fabricating the custom-made brain mold; Yifat Prut, Sofia Landi, Farid Aboharb for comments on a previous version of the manuscript. The authors acknowledge the Indiana University Pervasive Technology Institute for providing HPC resources that have contributed to the research results reported within this paper (URL: https://pti.iu.edu/). This work was supported by Leon Levy Fellowship to IS. The New York Stem Cell Foundation and, the NSF BCS-1057006 to WAF; NSF IIS-1636893, NSF BCS-1734853, NIH NIMH ULTTR001108, Indiana Clinical and Translational Institute (Passthrough), Microsoft Research Award - grant number Azure Credits Award, the Indiana University -grant number Areas of Emergent Research initiative 'Learning: Brains, Machines, Children', NIH 1U54MH091657, and Pervasive Technology Institute to FP. WAF is a New York Stem Cell Foundation - Robertson Investigator. This paper's content is solely the responsibility of the authors and does not necessarily represent the official views of the NIH or the National Science Foundation.

## Additional information

### Funding

| Funder | Grant reference number | Author |
|---|---|---|
| Leon Levy Foundation | | Ilaria Sani |
| New York Stem Cell Foundation | | Winrich A Freiwald |
| National Science Foundation | BCS-1057006 | Winrich A Freiwald |
| National Institute of Mental Health | ULTTR001108 | Franco Pestilli |
| Indiana Clinical and Translational Sciences Institute | Passthrough | Franco Pestilli |
| Microsoft Research | Azure Credits Award | Franco Pestilli |

| Indiana University | Areas of Emergent Research initiative Learning: Brains-Machines-Children | Franco Pestilli |
| --- | --- | --- |
| National Institutes of Health | 1U54MH091657 | Franco Pestilli |
| National Science Foundation | IIS-1636893 | Franco Pestilli |
| National Science Foundation | BCS-1734853 | Franco Pestilli |

The funders had no role in study design, data collection and interpretation, or the decision to submit the work for publication.

### Author contributions

Ilaria Sani, Conceptualization, Data curation, Formal analysis, Funding acquisition, Investigation, Visualization, Methodology, Writing—original draft, Project administration, Writing—review and editing; Brent C McPherson, Data curation, Formal analysis, Methodology; Heiko Stemmann, Conceptualization, Formal analysis; Franco Pestilli, Conceptualization, Resources, Software, Supervision, Funding acquisition, Writing—review and editing; Winrich A Freiwald, Conceptualization, Resources, Supervision, Funding acquisition, Writing—review and editing

### Author ORCIDs

Ilaria Sani http://orcid.org/0000-0002-4389-7263
Heiko Stemmann http://orcid.org/0000-0002-7492-1115
Franco Pestilli https://orcid.org/0000-0002-2469-0494
Winrich A Freiwald https://orcid.org/0000-0001-8456-5030

### Ethics

Animal experimentation: Our Protocols protocol has been approved by The Rockefeller University, Institutional Animal Care and Use Committee. In vivo imaging procedures were performed at the Center for Advanced Imaging of Bremen University. They conformed to the National Institutes of Health Guide for Use and Care of Laboratory Animals, regulations for the welfare of experimental animals issued by the federal government of Germany, and stipulations of local Bremen authorities

### Decision letter and Author response

Decision letter https://doi.org/10.7554/eLife.40520.022
Author response https://doi.org/10.7554/eLife.40520.023

## Additional files

### Supplementary files

• Supplementary file 1. Functional ROI mapping. For each functional ROI used during tracking procedure we report the behavioral task used for the functional mapping, the AP position of stereotactic coordinates in AC coordinate (center of ROI), and the anatomical AP position of the corresponding areas in *Saleem and Logothetis (2007)*.
DOI: https://doi.org/10.7554/eLife.40520.017
• Transparent reporting form
DOI: https://doi.org/10.7554/eLife.40520.018

### Data availability

All data generated or analysed during this study are included in the manuscript and supporting files. Source data for Figures 2, 3, 4, 5, have been made available via the Open Science Framework (https://osf.io/8ks5t/).

The following dataset was generated:

| Author(s) | Year | Dataset title | Dataset URL | Database and Identifier |
|---|---|---|---|---|
| Sani I, McPherson B | 2018 | Functionally defined white matter of the macaque monkey brain reveals a dorso-ventral attention network - Data | https://osf.io/8ks5t/ | Open Science Framework, 10.17605/osf.io/8ks5t |

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
