## [Decision Letter]

Thank you for submitting your article "A dorso-ventral attention network in the macaque monkey brain" for consideration by *eLife*. Your article has been reviewed by three peer reviewers, including Heidi Johansen-Berg as the Reviewing Editor and Reviewer #1, and the evaluation has been overseen by Joshua Gold as the Senior Editor. The following individual involved in review of your submission has agreed to reveal their identity: Igor Kagan (Reviewer #3). Reviewer #2 remains anonymous.

The reviewers have discussed the reviews with one another and the Reviewing Editor has drafted this decision to help you prepare a revised submission.

Summary:

This study addresses a fundamental and interesting question. It is an interesting follow up on the previous work from the same lab providing a functional description of area PITd.

The methods are strong and the data are high quality. The authors have done a good job of checking how robust their findings are to different potential methodological pitfalls.

The study uses ex-vivo high-resolution diffusion MRI (dMRI) to show that PITd is also strongly connected to dorsal attentional/oculomotor areas LIP and FEF. The first part of results elucidated direct PITd-LIP and PITd-FEF connectivity; the second part of results identified specific white matter fasciculi responsible for those connections. Taken together, these results are an important step towards understanding the attentional network, and will contribute substantially to the ongoing debate about similarities and differences in visuospatial and attentional circuitry in humans and monkeys.

Essential revisions:

1) The authors make a strong claim that some tracts mediate the connections between areas and others don't. However, tractography is not equally good at reconstructing each tract. For instance, one hypothesis in the current study was the arcuate fascicle (AF) would carry connections between FEF and temporal cortex (in line with early work by Petrides). Instead the authors find the extreme capsule (EmC) to carry these connections. Although entirely plausible, AF is much harder to reconstruct using tractography than EmC, especially in the macaque. Although the authors run a test to see whether they can reconstruct all tracts, it would be good to see a demonstration that the results are not driven by greater difficulty of finding AF than 'simple' longitudinal tracts such as SLF3 and EmC.

2) The authors claim that some connections belong to certain fiber bundles, but as far as I can see, this is only substantiated by visually inspecting overlap. Can the authors expand this analysis by formally quantifying the similarity in the tracts identified by running their ROI-to-ROI analysis with the reconstructed fiber bundles? This will also allow them to show overlap of each ROI-to-ROI analysis with each of the reconstructed tracts, providing a much better overview of the data;

3) Importantly, although the authors nicely show tracts between PITd and LIP and FEF, it is hard to judge their value without knowing the (lack of) connectivity of nearby regions. One would hope the results are quite specific. Although it is likely that regions nearby PITd share some of its connections, for instance we know that EmC projects to more anterior STS regions, the claim of the authors would be substantially strengthened if they can show that the pattern of connections (the "connectivity fingerprint", e.g., strong ILF and EmC, weak AF) is unique to PITd and that surrounding areas with a different function have a different connectional profile. This to me seems an important demonstration of the specificity of the result and would help ascribing a specific role to PITd.

4) The Introduction is well written and convincingly sets up the story, largely from the attentional circuitry (and more general) connectivity perspective. But one important facet is missing, in my opinion: a concise mentioning of the ventral attention network (in humans), and the concept about the dorsal/ventral attention network functional division. Please note that I am not suggesting that authors dive deep into ongoing debates on differences between dorsal and ventral network functionality, lateralization, and the specific function of the specific ventral attention network node, TPJ, in re-orienting the attention (Corbetta and Shulman, 2005, 2011) vs. contextual updating (Geng and Vossel, 2017) and even social cognition (Carter et al., 2013; Mars et al., 2013; Schwiedrzik et al., 2015). But at least a brief acknowledgement of the issue at hand seems to be in order. I also realize that authors do not consider the PITd the analog of human TPJ, and not a part of "ventral attention network" (if it at all exists in monkeys), but readers who did not read Stemmann and Freiwald, 2016 might initially think so, at least until they get to the Discussion.

---

## [Author Response]

Essential revisions:1) The authors make a strong claim that some tracts mediate the connections between areas and others don't. However, tractography is not equally good at reconstructing each tract. For instance, one hypothesis in the current study was the arcuate fascicle (AF) would carry connections between FEF and temporal cortex (in line with early work by Petrides). Instead the authors find the extreme capsule (EmC) to carry these connections. Although entirely plausible, AF is much harder to reconstruct using tractography than EmC, especially in the macaque. Although the authors run a test to see whether they can reconstruct all tracts, it would be good to see a demonstration that the results are not driven by greater difficulty of finding AF than 'simple' longitudinal tracts such as SLF3 and EmC.

We thank the reviewers for this great suggestion. To aid the comparison between different tracts, we now more clearly show the results of anatomical tracking for all the alternative hypotheses we tested. In addition, we calculated quantitative measures as an estimation for tracking difficulty. More specifically, we report the streamline density that each tract returned (Figure 4D,H,N), a measure that takes into account the number of streamlines obtained with tracking and corrects it for the ROI size. We found that none of the tracts that turned out to host attention streamlines were easier to track when compared to the alternative hypothesis. More specifically, density was similar for different tracts. For example, tracking EmC turned out to be as hard as tracking AF (Figure 4N). This is mostly because EmC necessitates exclusion masks to be confined to the narrow portion of white matter between the claustrum and the fundus of the lateral fissure, without going medial to the putamen.

We now report tracking difficulty in Figure 4, explain the details in Materials and methods, and comment on the results in the main text.

2) The authors claim that some connections belong to certain fiber bundles, but as far as I can see, this is only substantiated by visually inspecting overlap. Can the authors expand this analysis by formally quantifying the similarity in the tracts identified by running their ROI-to-ROI analysis with the reconstructed fiber bundles? This will also allow them to show overlap of each ROI-to-ROI analysis with each of the reconstructed tracts, providing a much better overview of the data;

We thank the reviewers for these helpful suggestions. In the previous version of the manuscript we, indeed, only visually inspected the overlap between our functional tracts and major anatomical pathways. We have now implemented an approach that formally quantifies the path similarity between functionally and anatomically identified tracts.

As detailed in Materials and methods, we calculated the percentage of overlap between the functionally defined “attention tracts” and the major anatomical pathways as follows: overlapping volume in the two tracts divided by the total volume of the attention tract. Results are reported in Figure 4C,G,M. They are consistent with our previous conclusions. We are convinced this analysis has improved the rigor and reproducibility of our results in evaluating alternative hypothesis about the anatomical pathways our functional tracts take.

3) Importantly, although the authors nicely show tracts between PITd and LIP and FEF, it is hard to judge their value without knowing the (lack of) connectivity of nearby regions. One would hope the results are quite specific. Although it is likely that regions nearby PITd share some of its connections, for instance we know that EmC projects to more anterior STS regions, the claim of the authors would be substantially strengthened if they can show that the pattern of connections (the "connectivity fingerprint", e.g., strong ILF and EmC, weak AF) is unique to PITd and that surrounding areas with a different function have a different connectional profile. This to me seems an important demonstration of the specificity of the result and would help ascribing a specific role to PITd.

We thank the reviewers for making this important request on the specificity of connections. In response to this request we performed extensive new analyses in which we functionally and anatomically parceled areas nearby PITd and analyzed the connectivity of those regions with LIP and FEF. More specifically, we defined MST, MT, and FST by means of a motion localizer, V4t by means of attentional modulation, PL by using a face localizer, and TEa anatomically (Figure 4—figure supplement 4, panels A,D). Then, we tracked the connections between these areas and functionally defined target attention areas LIP and FEF. Finally, we analyzed the connectivity fingerprint for each area by calculating the percentage of overlap with traditional anatomical pathways (Figure 4—figure supplement 4, panels B,C,E,F).

Results show that (1) PITd connectivity with LIP is specific in the sense of a strong vILF focus (Figure 4—figure supplement 4, panel B). The spatial profiles of PITd and neighboring areas are, in general, similar with most of the overlap occurring for vILF. The main exceptions to this rule are areas PL which appears to lack direct connections to LIP, to a lesser extent, MST which shown a minimal overlap with both vILF and VOF, and V4t whose connectivity with LIP is less confined to vILF than PITd’s. (2) For STS to FEF connectivity, differences in connectivity exist primarily between groups of areas lying in the fundus of the STS, in the lower bank of the STS, and in the inferior temporal gyrus (Figure 4—figure supplement 4, panels E). Here, motion sensitive areas MST, MT, and FST showed less degree of overlap with ILF and EmC as compared to V4t, PITd, TEa; PL appears to lack direct connections to FEF. To some extent, MST represents an exception, as it shows almost the same degree of overlap with ILF-EmC and AF.

These new results provide important new insights. They confirm the existence of tract-specific connections from PITd and demonstrate that this connectivity is part of a larger-scale pattern of connectivity that PITd shares with its immediate neighborhood. Thus, functional specificity of PITd does not arise as a simple consequence of its connectivity with LIP and FEF. PITd is fundamentally different from nearby areas in not being motion-selective an in being very strongly attention modulated, yet the pattern of connectivity with LIP and FEF is largely shared. This is an important insight and one that makes this network different from the nearby face-processing network. Whereas in the face-processing network connectivity ensures domain-specificity (for faces), it cannot do so in the case of attention areas which require connections to a multitude of sensory areas to exert their cognitive, domain-general functions. Yet the combination of function and connectivity really defines both as bona-fide networks. The second main insight is that none of the white-matter bundles supporting LIP-FEF-PITd connectivity, are attention-specific functional bundles.

We added these new results in the main text, show them as a supplementary figure, and comment them in the Discussion.

4) The Introduction is well written and convincingly sets up the story, largely from the attentional circuitry (and more general) connectivity perspective. But one important facet is missing, in my opinion: a concise mentioning of the ventral attention network (in humans), and the concept about the dorsal/ventral attention network functional division. Please note that I am not suggesting that authors dive deep into ongoing debates on differences between dorsal and ventral network functionality, lateralization, and the specific function of the specific ventral attention network node, TPJ, in re-orienting the attention (Corbetta and Shulman, 2005, 2011) vs. contextual updating (Geng and Vossel, 2017) and even social cognition (Carter et al., 2013; Mars et al., 2013; Schwiedrzik et al., 2015). But at least a brief acknowledgement of the issue at hand seems to be in order. I also realize that authors do not consider the PITd the analog of human TPJ, and not a part of "ventral attention network" (if it at all exists in monkeys), but readers who did not read Stemmann and Freiwald, 2016 might initially think so, at least until they get to the Discussion.

The reviewers are right. This is a major oversight on our part. We regret not including contributions from human studies on dorsal/ventral attention network functional division. We have now incorporated contributions about the human ventral attention network and the various roles it has been proposed to play, along with a clear view on PITd, not showing typical features of ventral attention areas described in humans.